# Medium-Term Clinical Outcomes of the Global Icon Stemless Shoulder System: Results of a 2-Year Follow-Up

**DOI:** 10.3390/jcm12216745

**Published:** 2023-10-25

**Authors:** Robert Zbeda, Raf Asaid, Sarah A. Warby, John Tristan Cassidy, Gregory Hoy

**Affiliations:** 1Melbourne Orthopaedic Group, 33 The Avenue, Windsor, VIC 3181, Australia; robert.zbeda@rothmanortho.com (R.Z.); raf.asaid@yahoo.com.au (R.A.); tristancassidy@gmail.com (J.T.C.); gahoy@mog.com.au (G.H.); 2Melbourne Shoulder Group, 305 High Street, Prahran, VIC 3181, Australia; 3Department of Physiotherapy, Podiatry, Prosthetics and Orthotics, La Trobe University, Corner of Kingsbury Drive and Plenty Road, Bundoora, VIC 3080, Australia; 4Department of Surgery, Monash Medical Centre, Monash University, Level 5, Block E·246 Clayton Road, Clayton, VIC 3168, Australia

**Keywords:** Global Icon, stemless, arthroplasty, shoulder, outcomes, surgery

## Abstract

The primary aim of this study was to assess the medium-term outcomes of the Global Icon stemless shoulder replacement in patients who have undergone primary total shoulder arthroplasty (TSA) for glenohumeral joint osteoarthritis. A retrospective review of patients who had undergone a TSA using the Global Icon stemless shoulder system was performed. The Western Ontario Osteoarthritis Shoulder (WOOS) Index and Oxford Shoulder Score (OSS) were evaluated pre-operatively and at 12 to 24 months post-operatively. Radiological outcomes, operation time, and post-operative complications were reported. Primary analysis for the WOOS Index and OSS focused on detecting within-group treatment effects at 24 months using a repeated measures ANOVA. Thirty patients were included in the study. Post-surgery, there was a significant improvement at 24 months on the OSS (ES = 0.932, CI: 41.7 to 47.7, *p* < 0.001) and the WOOS Index (ES = 0.906, CI: 71.9 to 99.8, *p* < 0.001). Radiographs revealed that no component loosened, migrated, or subsided. The median operative time was 75.5 (IQR: 12.25, range: 18 to 105) min. No implant-related complications were reported. The Global Icon stemless replacements have excellent clinical outcomes in this cohort at 12- and 24-month follow-up with no implant-related complications.

## 1. Introduction

Glenohumeral joint osteoarthritis is a loss of the articular surface of the glenoid and the humeral head and typically results in pain and degradation of upper limb function [1]. It affects up to 32.8% of people in the United States of America over the age of 60 years [1], and with an increasingly aging population worldwide, it is and will continue to be a disease burden for a growing proportion of the people in many countries.

Anatomic total shoulder arthroplasty (TSA) is a successful treatment option for glenohumeral osteoarthritis with an intact rotator cuff [2]. There has been a trend towards bone-preserving humeral components in anatomic total shoulder replacement [3]. Stemless shoulder implants, which lack a conventional diaphyseal humeral stem and rely on metaphyseal fixation, have been recently introduced to avoid stem-related complications of the stemmed shoulder implant. The stemless prosthesis has been suggested to yield several advantages including improved bone preservation, decreased stress shielding, shorter operative time, and easier removal at revision [4,5,6]. These theoretical advantages have contributed to the current popularity of stemless total shoulder replacement but have yet to be fully supported by quality comparison studies.

The biomechanical basis for the success of stemless implants has been confirmed [3,7], and clinical studies have consistently demonstrated a significant improvement in shoulder function post-operatively with a low risk of implant-related complications [8,9,10,11,12]. A recent systematic review [13] reported excellent medium-term outcomes of the stemless implants, comparable to reports on its stemmed counterpart.

The Global Icon (Depuy Synthes, Raynham, MA, USA) stemless shoulder system is relatively new and has been approved in Europe and Canada since April 2017 and September 2017, respectively. To date, there has only been one study investigating the short-term outcomes of the stemless Global Icon. Smith et al. [14] reported a significant improvement on the Constant-Murley Score (CMS), Oxford Shoulder Score (OSS) and EQ-5D-5L at 12-month follow-up on 154 patients. Complications were reported in 22 of the 154 participants; however, only 2 required revision surgery.

Therefore, the aim of this study is to present medium-term clinical outcomes of 30 total shoulder replacements (TSRs) implanted for glenohumeral osteoarthritis using the Global Icon stemless shoulder system, with a follow-up period of 24 months.

## 2. Materials and Methods

We performed a single surgeon, retrospective review of a consecutive series of patients from a single institution (The Melbourne Orthopaedic Group (33 The Avenue, Windsor, VIC 3181, Australia)) undergoing primary TSA. Ethical approval for this study was obtained from Ramsay Health Care (Ethical Approval 2022-002-LNR).

### 2.1. Selection Criteria

Patients were included if they had undergone primary anatomic TSA using the Global Icon between January 2018 and February 2020 for glenohumeral joint osteoarthritis. Patients with pre-operative inflammatory joint disease, avascular necrosis, non-functioning rotator cuff, poor metaphyseal bone stock, acute trauma, and infection were excluded. Patients with no follow-up patient reported outcome measures (PROMs) were also excluded. Early indications to use stemless instead of stemmed component were deformity of humeral shape. The previously reported success of the stemless component [8] made it the prothesis of choice to minimise humeral shaft bone exposure in case of later revision.

### 2.2. Pre-Operative Work-Up

All patients were assessed medically for anaesthetic suitability. A pre-operative CT scan was performed with DICOM views sent for “Trumatch” computer planning and sizing. Three-dimensional models were not made as PSI was not utilized in this group. Contralateral imaging confirmed sizing in unusual pathologies. A pre-operative haematolological workup was performed but no blood was cross-matched as no patient was transfused by the senior author in primary arthroplasty.

### 2.3. Operative Procedure

All surgeries were performed by 1 surgeon (GH) with the patient in the semi-supine position with a pneumatic arm holder under regional nerve block. A deltopectoral approach and subscapularis tenotomy was performed. A guided anatomic resection of the humeral head was made with the ring of the guide resting on the posterior and superior cuff insertions, while being aligned with the anatomic neck to achieve the correct inclination and retroversion. An Anchor Peg Glenoid (DePuy Synthes, Warsaw, IN, USA) was implanted with cement in the peripheral holes only for immediate fixation (and none on the under surface of the component), and cancellous bone autograft reamings were applied circumferentially around the central interference peg for bony ongrowth, as previously published in short and longer term outcome studies using this implant [15,16] (Figure 1). The Global Icon humeral head was sized off the resected humeral head and implanted with the base plate gaining excellent metaphyseal bone contact and morse taper connection to the head. The B glenoids had high-sided reaming and no structural bone grafts. The rationale for this has been previously published [17].

The subscapularis tenotomy was subsequently repaired end-to-end with a contracting strong, synthetic suture in an interrupted fashion (Dynacord^®^, Warsaw, IN, USA). A standardized rehabilitation protocol was utilized with shoulder pendulum exercises in the first 2 weeks after surgery. The sling was discarded at 4 weeks, and the patient then began progressive active assisted range of motion exercises at 2 weeks post-operatively and started strengthening exercises at 6 weeks post-operatively. Unlimited physical activities were allowed at 3 months.

### 2.4. Outcome Measures

The primary outcome measures included the Western Ontario Osteoarthritis Shoulder (WOOS) Index and the OSS. Patients were asked to complete these scores pre-operatively and 12 to 24 months post-operatively via a secure online platform. The WOOS Index is a self-administered tool with 19 items over the four domains of physical symptoms, sport/recreation/work, lifestyle function, and emotional function. The total score is expressed as a percentage with 100% representing a normal, healthy shoulder. The WOOS Index is valid, reliable, and responsive for measuring change in patients with shoulder osteoarthritis [18]. The OSS is a self-administered tool, with 12 items assessing shoulder pain and function. The total score for the OSS can range between 0 and 48 points where 48 represents no deficit. The OSS is a valid and reliable measure of general shoulder function [19]. To date, no studies have reported a minimally clinically important difference (MCID) value for the WOOS Index, while the MCID for the OSS is 6.3 points [20].

Radiographs were evaluated for radiolucency, osteolysis, loosening, migration, fractures, glenohumeral subluxation, and subsidence by an independent fellowship-trained orthopaedic surgeon at routine follow-up. Radiological evaluation of the glenoid component for component loosening was graded as described in Lazarus et al. [21]. Analysis of the humeral component for radiolucent lines was performed using the zones on anteroposterior (zones 1–5) and axillary (zones 6–10) radiographs as outlined by Smith et al. [14].

Operative time and the occurrence of any post-operative compilations (including revision surgery) were recorded. Operative time reflects the time from incision to wound closure [22]. Implant-specific complications were defined as complications caused by the defining characteristics of stemless shoulder replacements compared to conventional stemmed anatomic shoulder replacement.

### 2.5. Statistical Analysis

Analysis was performed using SPSS (version 28.0, IBM, New York, NY, USA). Primary analysis for the WOOS Index and the OSS focused on detecting within-group treatment effects at the 24-month time point (with effect sizes and 95% CIs) using a repeated measures ANOVA. Missing data for PROMs were handled by using ANOVA with listwise deletion [23]. Post-hoc analysis was performed (repeated measures ANOVA pairwise comparisons) to investigate the differences between each time point: baseline to 12 months, 12 to 24 months, and baseline to 24 months. Paired samples *t* tests were performed to obtain a t statistic and effect size (Cohen’s d) for pairwise comparisons. An effect size of 0.2 to 0.5 would be considered a small effect, 0.2 to 0.5 a medium effect, and >0.8 a large effect [24]. Radiological outcomes, operation time, and post-operative complications were reported with descriptive statistics.

## 3. Results

### 3.1. Patient Characteristics

The practice database retrieved 43 participants who had undergone a total shoulder replacement using the Global Icon stemless system. Thirteen participants had no follow-up PROMs, leaving a total of 30 participants included in the study (Figure 2). Glenoid type for the cohort was as follows: type A1: six participants, type A2: eleven participants, type B1: three participants, type B2: five participants, and type C: four participants.

Operations were performed by one consultant shoulder surgeon (GH). All components were implanted in conjunction with a glenoid anchorpeg component as previously published [16,17] as part of a TSA. There were 11 men (37%) and 19 women (63%) with a mean age of 69 years and age range between 59 and 79 years. Average in-patient stay was 3 days. No patient required any blood products.

### 3.2. Outcomes

#### 3.2.1. Patient Reported Outcomes

Post-surgery, there was a significant improvement at 24 months on the OSS (ES = 0.932, CI: 41.7 to 47.7, *p* < 0.001, Figure 3, Table 1) and the WOOS Index (ES = 0.906, CI: 71.9 to 99.8. *p* < 0.001, Figure 4, Table 1). There was a significant improvement for all sub-sections of the WOOS Index (physical symptoms, sport/recreation/work, lifestyle, and emotions) at 24 months (*p* < 0.001) (Appendix A). Post-hoc analysis (Table 2) revealed a significant improvement between baseline and 12 months for the OSS (ES = 6.9, CI: 12.8 to 23.4, *p* < 0.001) and the WOSS Index (ES = 16.4 CI: 36 to 63.8, *p* < 0.001), and baseline and 24 months for the OSS (ES = 5.4, CI = 15.6 to 23.7, *p* < 0.001) and the WOOS Index (ES = 16.5, CI: 40.7 to 64.5, *p* < 0.001). There was no significant difference between 12- and 24-month scores for the OSS and the WOOS Index, indicating that most improvements occurred in the first 12 months post-surgery.

#### 3.2.2. Radiological Outcomes

Radiological outcomes are outlined in Table 3. Twenty-six participants had radiographs available for review at a mean follow-up of 64 weeks. No component significantly loosened, migrated, or subsided. All glenoid components were grade 0. Ultimately, 21 of the 26 participants and 17 of 26 participants had anteroposterior and axillary views respectively available for review. For the humeral component, radiolucencies were minimal and located in zones 1, 5, 6, and 10. Figure 5 displays an anterior to posterior post-operative X-ray, and Figure 6 displays an axillary lateral post-operative X-ray showing the anchorpeg position.

#### 3.2.3. Complications and Operative Time

No implant-related complications were recorded. One patient had a transient partial posterior cord plexopathy resolving within 6 weeks and thought to be related to the nerve block. One patient has symptoms from a possible supraspinatus cuff failure and is being monitored but has not been revised. There were no episodes of infection, instability, or unexplained pain. The median operative time was 75.5 min (IQR: 12.25, range:18 to 105, mean: 78.6 min, SD: 15.5).

## 4. Discussion

This study demonstrates good medium-term results of the Global Icon stemless TSA.

There were significant improvements on the OSS and the WOOS Index post-surgery at 12 and 24 months compared to pre-operative scores. The size of the treatment effect at 24 months was above 0.8 for both outcomes, indicating patients’ perceived effectiveness of treatment was great [24]. In addition, all subsections of the WOOS Index showed significant improvements at 24 months, with effect sizes over 0.8, indicating that patients’ perceived change in health status in the domains of physical symptoms, sport and recreation, lifestyle, and emotions was also large [18,24]. To date, no studies have reported a minimally clinically important difference (MCID) value for the WOOS Index, while the MCID for the OSS is 6.3 points [20]. Given that the mean difference on the OSS at 12 and 24 months exceeded the MCID, the results on the OSS indicate not only a statistically significant difference but also a clinically important change post-surgery. Second, our mean operative time was 78.6 min (median: 75.5), which is considerably lower than operative times reported for other arthroplasty cohorts, which range from a mean of 221.6 to 313.5 min [22]. Lastly, there were no implant-related complications or radiological issues such as aseptic loosening or implant malposition reported.

Clinical studies investigating stemless systems alone have consistently demonstrated significant improvements in shoulder function with a low risk of implant-related complications [8,9,10,11,12]. Using the TESS system, Huguet et al. [10] published the first results of stemless implants in 63 patients at a mean follow-up of 3 years with significant improvements in CMS from 29.6 to 75.5. The authors reported five minor cracks in the lateral humeral cortex noticed on the first post-operative radiograph that all healed within 2 months without any change in the implant position. Using the Eclipse stemless in 49 shoulders, Hawi et al. [9] found a significant improvement in CMS (52% to 79%), shoulder flexion (101° to 118°), abduction (79° to 105°), and external rotation range of motion (21° to 43°) at 9-year follow-up. The humeral-sided complication rate was 9.3% with no implant-specific complications. Moursy et al. [11] found significant improvements post-operatively for range of motion and CMS with the Eclipse stemless at a mean follow-up of 7.5 years with no reported complications. Athwal et al. [25] found significant improvements in a range of clinical outcomes (ASES, WOOS Index, SF-12) with the Sidus implant at 6 months and at 1- and 2-year follow-up with a low complication rate. More recently, Ambros et al. [8] reported on 53 cases using the Eclipse stemless at a mean follow-up of 70 months. Post-operatively there was a significant improvement in the CMS from 53.8 to 83.5, and the overall complication rate was 18.9% (10/53). Complications included one transient upper trunk brachial plexus palsy that did not require revision surgery and one subscapularis insufficiency with chronic anterior subluxation of the glenohumeral joint that refused revision surgery. There were eight surgical revisions (15.7%) including one periprosthetic infection that led to a component change, one humeral head change to a larger size due to reoccurring subluxations, two because of glenoid wear, and one arthroscopic arthrolysis due to subacromial capsular fibrosis. Five patients underwent revision for rotator cuff and biceps tears. To date, there has only been one other study investigating short-term outcomes of Global Icon. Smith et al. [14] reported a significant improvement on the Constant-Murley, OSS, and EQ-5D-5L at 12-month follow-up on 154 patients. Complications were reported in 22 of the 154 participants; however, only 2 required revision surgery. Our results on the Global Icon are comparable to the aforementioned studies with a 2-year follow-up and no complications reported.

In studies directly comparing the stemmed vs. stemless component, the stemless component does not appear to perform inferiorly to its stemmed counterpart and may offer several advantages including reduced operating time and reduced blood loss. In a retrospective review, Berth et al. [26] found no difference in Constant scores between a stemmed and stemless prothesis group at 32 months follow-up; however, the estimated blood loss and mean operating time were significantly lower in the stemless group. Similarly, Heuberer et al. [27] found operating times were significantly lower in their stemless group compared to the stemmed patient group. Randomised controlled trials have found no significant difference in PROMs [12,28,29], range of motion [12,29], and revision rate [12] of stemmed vs. stemless implants. Given the similar clinical outcomes of both types of implants, the stemless prothesis may be the preferred choice due to its advantages of reduced operating time and blood loss [26].

Complication rates of stemless implants have been comparable to stemmed implants. In a systematic review, Upfill-Brown et al. [13] found a 0.2% rate of asymptomatic humeral loosening (none of which required revision) and a 0.5% rate of intraoperative humeral fracture in patients undergoing stemless TSA or hemiarthroplasty [13]. In a systematic review of anatomic and reverse stemmed implants, Bohsali et al. [30] reported a 0.1% rate of humeral loosening and 0.6% rate of intra-operative fracture. Humeral radiolucencies and bone mineral density decrease have been reported in multiple studies following stemless arthroplasty [9,25,27,31]. The rates of humeral radiolucencies post-operatively have ranged anywhere from 2% to 37%, but no influence on clinical outcomes or humeral related revisions have been reported [13,27].

The stemless implant may have advantages over the stemmed system when it comes to pre-operative planning with three-dimensional (3D) templating software. Freehill and colleagues [32] evaluated the accuracy of implant size selection based on 3D templating using both stemmed and stemless implants. The authors found that virtual pre-operative planning can reliably predict the component size utilized for anatomic total shoulder arthroplasty, though the stemless system had a greater predicted accuracy for the humeral component (98% matched the pre-operative template within one size with 79% exactly matched) when compared to the stemmed system (88% of cases were within one size of the preoperative plan with 83% exactly matched). The accuracy of the humeral head sizes on the stemmed component was also overall lower with 79% matching within one size compared to 97% with a stemless humeral implant. Statistical differences between stemmed and stemmed groups for pre-operative 3D accuracy were not analysed. Enhanced accuracy of 3D pre-operative planning may reduce implant costs, operative times, and inventory processes related to shoulder arthroplasty [32]. More research is required to determine the effect of accurate pre-operative 3D technologies on clinical outcomes in shoulder arthroplasty, both for stemmed and stemless systems [33].

Other potential advantages of the stemless implant have been reported in the literature, including a lack of disruption of the endosteal blood supply, improved accuracy in sizing the humeral head, preservation of bone stock for future revisions, reduced risk of intra- and post-operative periprosthetic humeral shaft fractures, elimination of broaching, replacement of the humeral head independently from the humeral axis in post-traumatic deformities, and reduced stress shielding [13,25,34]. In addition, a problematic stemless implant may be more easily revised to a standard stemmed implant than a standard stemmed implant revised into a long-stem implant [25]. Potential disadvantages of the stemless systems include difficult positioning of the humeral component resulting in non-anatomic placement, early wear, risk of component loosening before osseous ingrowth, and reliance on adequate humeral metaphyseal bone stock [13]. Planned [35] and future randomised trials of stemmed vs. stemless protheses will be required to determine whether some of these theoretical advantages and disadvantages are in fact real clinical entities.

A strength of this study is that it is the first to investigate outcomes for the Global Icon with medium-term follow-up (24 months), showing a significant improvement in shoulder function comparable to other clinical outcomes in stemless and stemmed protheses. In addition, this study used a PROM that is sensitive and specific to measuring change in the glenohumeral joint osteoarthritis population (the WOOS Index), in contrast to many studies which do not [14,26]. Our lack of complications, short operating time, nil transfusions, and pathology-specific outcome measure make this a good surgical option for anatomic shoulder arthroplasty.

Limitations of our study include the retrospective nature, small number of included patients, and lack of randomized control or comparison groups, although we have previously studied the outcomes of the same glenoid component in published short- and medium-term outcome studies [15,16]. In addition, of the 43 Global Icon patients retrieved in our database, 30 had outcomes available for analysis resulting in possible measurement error. We acknowledge additional follow-up will be needed to determine the long-term outcomes of these patients.

## 5. Conclusions

This study demonstrates significant improvements in shoulder function after treatment of shoulder osteoarthritis with the Global Icon stemless system. Our study on the Global Icon system is at least comparable if not favourable when compared to other published studies on stemless implants, which demonstrate excellent medium-term outcomes and a low complication risk.

## Figures and Tables

**Figure 1 jcm-12-06745-f001:**
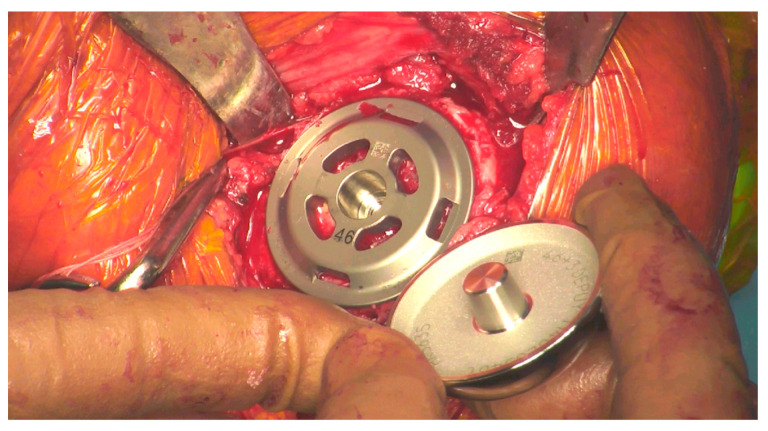
Head about to be impacted onto the stemless baseplate during surgery.

**Figure 2 jcm-12-06745-f002:**
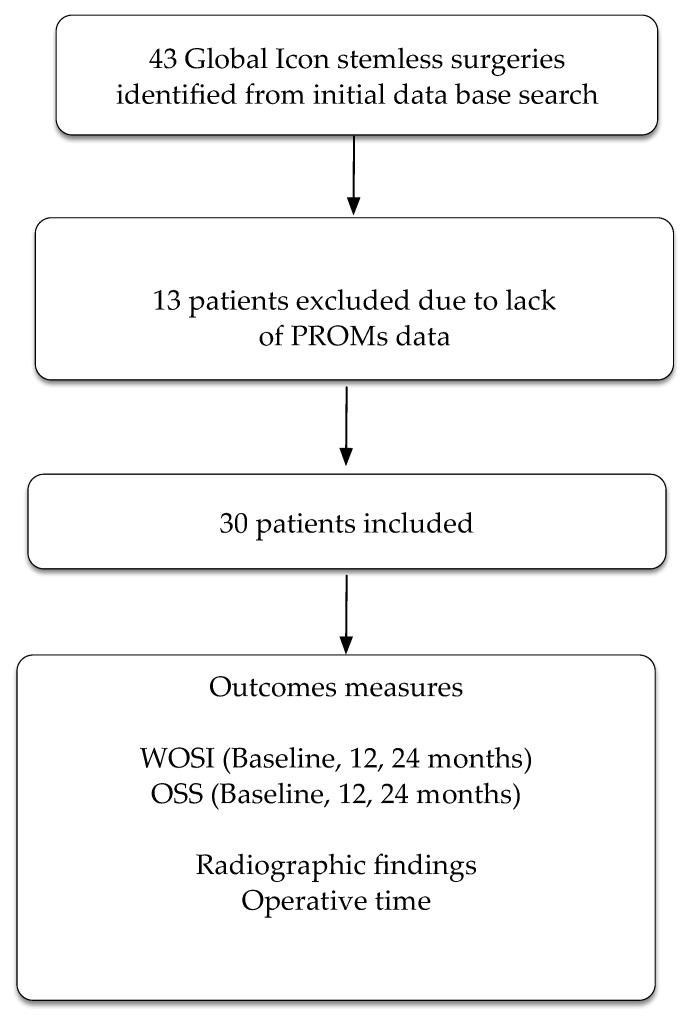
Flow chart of patients through the trial. PROMs = patient reported outcome measures. WOSI = Western Ontario Shoulder Index. OSS = Oxford Shoulder Score.

**Figure 3 jcm-12-06745-f003:**
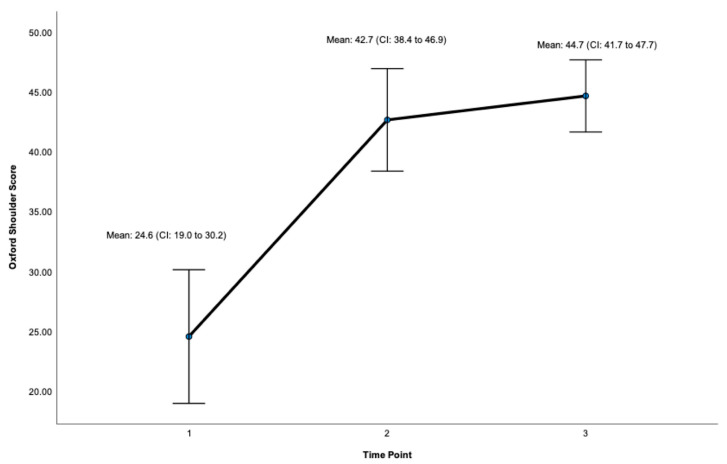
Total Score for the Oxford Shoulder Score (OSS). Timepoint 1 = Baseline, 2 = 12 months, 3 = 24 months. CI = 95% Confidence Interval.

**Figure 4 jcm-12-06745-f004:**
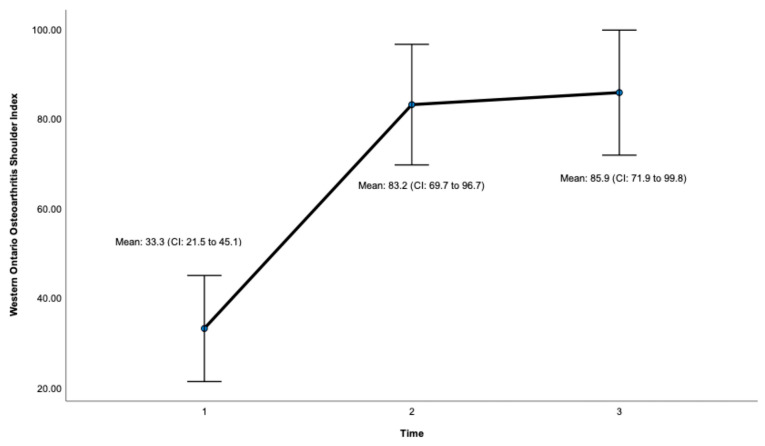
Total score for the Western Ontario Osteoarthritis Shoulder (WOOS) Index. Timepoint 1 = Baseline, 2 = 12 months, 3 = 24 months. CI = 95% confidence interval.

**Figure 5 jcm-12-06745-f005:**
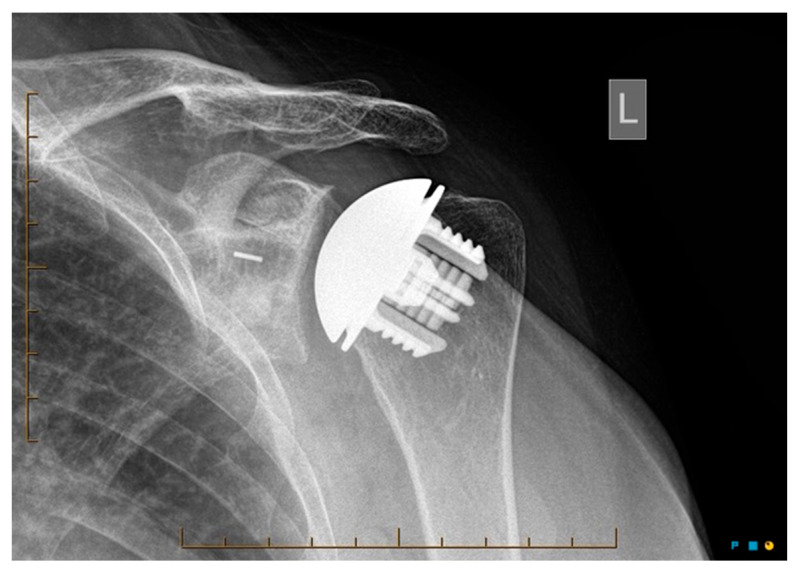
Anteroposterior post-op X-ray showing no overstuffing of the joint, 24-month post-operative. Female participant aged 62 years.

**Figure 6 jcm-12-06745-f006:**
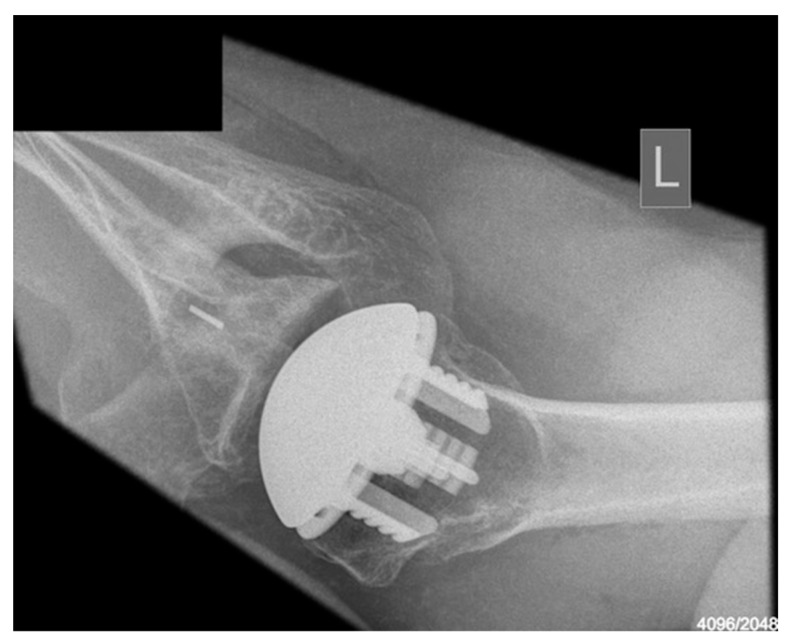
Axillary-lateral post-op X-ray showing anchorpeg position, 24-month post-operative. Female participant aged 60 years.

**Table 1 jcm-12-06745-t001:** Primary results for patient-reported outcomes.

	Mean (SD)	St. Error	95% CI	F Score	Effect Size (Partial Eta)	*p*-Value (ANOVA)
Oxford Shoulder Score						
Baseline	24.6 (8.8)	2.5	19.0 to 30.2			
12 months	42.7 (6.7)	1.9	38.4 to 46.9			
24 months	44.7 (4.7)	1.4	41.7 to 47.7	151.7	0.932	**<0.001**
WOOS Index (Total score)						
Baseline	33.3 (17.6)	5.3	21.5 to 45.1			
12 months	83.2 (20)	6.0	69.7 to 96.7			
24 months	85.9 (20.8)	6.3	71.9 to 99.8	96.9	0.906	**<0.001**

ES = Effect Size (Cohen’s d). WOOS Index = Western Ontario Osteoarthritis Shoulder Index. Significance set at 0.05. *p* values in bold indicate statistical significance.

**Table 2 jcm-12-06745-t002:** Post hoc analysis for patient-reported outcomes.

	Time Point	Comparisons	Mean Diff (St Error)	ES (95% CI)	t Value	*p* Value (ANOVA)
Oxford Shoulder Score	Baseline	12 months	18.1 (2.4)	6.9 (12.8 to 23.4)	−15.1	**<0.001**
		24 months	20.1 (1.6)	5.4 (16.5 to 23.7)	−13.3	**<0.001**
	12 months	24 months	2.0 (1.8)	6.3 (−2.0 to 6.0)	−1.1	0.30
WOOS (Total Score)	Baseline	12 months	49.9 (6.2)	16.4 (36.0 to 63.8)	−16.1	**<0.001**
		24 months	52.6 (5.3)	16.5 (40.7 to 64.5)	−11.3	**<0.001**
	12 months	24 months	2.7 (6.1)	20.2 (−10.0 to 16.3)	−0.4	0.67

ES = Effect Size (Cohen’s d). WOOS Index = Western Ontario Osteoarthritis Shoulder Index. Significance set at 0.05. *p* values in bold indicate statistical significance.

**Table 3 jcm-12-06745-t003:** Radiological outcomes.

				Humeral Head							Glenoid
	Anteroposterior Views						Axillary Views				
Zones	1	2	3	4	5	6	7	8	9	10	
Total (*n*)	21	21	21	21	21	17	17	17	17	17	25
Radiolucency	1	0	0	0	1	1	0	0	0	1	0
No radiolucency	20	21	21	21	20	16	17	17	17	16	25
	(95%)	(100%)	(100%)	(100%)	(95%)	(94%)	(100%)	(100%)	(100%)	(94%)	(100%)

Note. Grade 0 = no radiolucency. Grade 1 = Incomplete radiolucency around one or two pegs. Grade 2 = Complete radiolucency (approx. 2 mm wide) around one peg only, with or without incomplete radiolucency around one other peg. Grade 3 = Complete radiolucency (approx. 2 mm wide) around two or more pegs. Grade 4 = Complete radiolucency (>2 mm wide) around two or more pegs. Grade 5 = gross loosening.

## Data Availability

The data presented in this study are available on request from the corresponding author. The data are not publicly available due to privacy restrictions.

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
