# Peer review of "Medium-Term Clinical Outcomes of the Global Icon Stemless Shoulder System: Results of a 2-Year Follow-Up"

_jcm, 2023, doi:10.3390/jcm12216745_

Round 1

Reviewer 1 Report

The research aim is to analyze the outcomes of a TSA stemless implant in patients with shoulder OA. The subject is of interest to the readers.

The abstract is written and structured appropriately.

The introduction transposes the research into the topic and formulates the objective of the study at the end.

In the methodology section, the stages of the research are presented, and the results are clearly described.  As a general rule of reporting the p values: if p value is greater than 0.05 should be reported with two decimal values, if p value is between 0.001 and 0.05 should be reported with three decimal places and if values shown on output as 0.000 should be reported as <0.0001; I suggest to make this correction throughout the paper as p=0.000 is reported by authors multiple times throughout the paper.

The discussions interpret the research results and relate them to other results from scientific literature. Limitations of the study are provided at the end of the section.

The conclusions are concise and clear. Minor editing error – line 306 (the citations are written in superscript).

Editing recommendation – the citation should be before the end sentence dot, not after.

The references are adequate.

Author Response

Reviewer 1

The research aim is to analyze the outcomes of a TSA stemless implant in patients with shoulder OA. The subject is of interest to the readers.

The abstract is written and structured appropriately.

The introduction transposes the research into the topic and formulates the objective of the study at the end.

In the methodology section, the stages of the research are presented, and the results are clearly described.  As a general rule of reporting the p values: if p value is greater than 0.05 should be reported with two decimal values, if p value is between 0.001 and 0.05 should be reported with three decimal places and if values shown on output as 0.000 should be reported as <0.0001; I suggest to make this correction throughout the paper as p=0.000 is reported by authors multiple times throughout the paper.

The authors thank the reviewer for this comment.  We have now amended all text and tables (including the supplementary file) to show p values as recommended above.

The discussions interpret the research results and relate them to other results from scientific literature. Limitations of the study are provided at the end of the section. The conclusions are concise and clear.

Minor editing error – line 306 (the citations are written in superscript).

The authors thank the reviewer for this comment.  We have now amended these references, so they match the other references in the paper.

Editing recommendation – the citation should be before the end sentence dot, not after.

The authors thank the reviewer for this comment.  We have now placed references before the punctuation.

The references are adequate.

Reviewer 2 Report

The authors asses the outcomes (at 12 and 24 months) of primary total shoulder arthroplasty using the Global Icon stemless shoulder replacement. This kind of implant seems to have several advantages over the  stemmed implant so it is the preferred system nowadays. However, since it is a relatively new, there are only a few studies on this matter The study is well planned and well conducted. The cohort is relatively small. 

Author Response

Reviewer 2

The authors asses the outcomes (at 12 and 24 months) of primary total shoulder arthroplasty using the Global Icon stemless shoulder replacement. This kind of implant seems to have several advantages over the stemmed implant, so it is the preferred system nowadays. However, since it is a relatively new, there are only a few studies on this matter The study is well planned and well conducted. The cohort is relatively small.

The authors thank the reviewer for this comment.  We have acknowledged in the limitations that the cohort is relatively small. No edits are required based on this review.

Reviewer 3 Report

The objective of this paper is to assess the clinical outcomes for elective cases  of a stemless total shoulder implant with a 24 month follow-up.

Minor editing error in the abstract – line 24 after 99.8 there is an extra dot.

The introduction offers sufficient background on the theme and the objective of the study is formulated at the end.

The study design and statistical analysis are appropriate and well conducted. At the “Selection Criteria” subsection a flow chart with the studied population should be provided.

In the results section the only observation is with the p-value: although the statistical software offers p values equal to 0.000 as output, they should be reported in text with p<0.0001.

In the discussion section information on the use of the 3d-technologies to assess stemless vs stemmed shoulder arthroplasties in relation to other scientific papers would enhance this part of the paper, for e.g. for e.g. Moldovan, F.; Gligor, A.; Bataga, T. Structured Integration and Alignment Algorithm: A Tool for Personalized Surgical Treatment of Tibial Plateau Fractures. J. Pers. Med. 2021, 11, 190. doi: 10.3390/jpm11030190

The conclusion supports the study.

The bibliography  is appropriate but can be extended as suggested above.

Author Response

Reviewer 3

The objective of this paper is to assess the clinical outcomes for elective cases  of a stemless total shoulder implant with a 24 month follow-up.

Minor editing error in the abstract – line 24 after 99.8 there is an extra dot.

The authors thank the reviewer for this comment.  We have amended this to a comma as per the other results.

The introduction offers sufficient background on the theme and the objective of the study is formulated at the end.

The study design and statistical analysis are appropriate and well conducted. At the “Selection Criteria” subsection a flow chart with the studied population should be provided.

We thank the reviewer for this comment.  Please see new Figure 2 (Line 164) Flow Chart placed in the manuscript.

In the results section the only observation is with the p-value: although the statistical software offers p values equal to 0.000 as output, they should be reported in text with p<0.0001.

The authors thank the reviewer for this comment.  We have now amended all text and tables (including the supplementary file) to show p values as recommended above.

In the discussion section information on the use of the 3d-technologies to assess stemless vs stemmed shoulder arthroplasties in relation to other scientific papers would enhance this part of the paper, for e.g. for e.g. Moldovan, F.; Gligor, A.; Bataga, T. Structured Integration and Alignment Algorithm: A Tool for Personalized Surgical Treatment of Tibial Plateau Fractures. J. Pers. Med. 2021, 11, 190. doi: 10.3390/jpm11030190

We thank the reviewer for this comment.  We have added the following paragraph to the discussion in regards to 3D technologies, however we have used references that are shoulder arthroplasty specific rather than the one mentioned above for tibial plateau fractures.

Line 309

The stemless implant may have advantages over the stemmed system when it comes to pre-operative planning with three-dimensional (3-D) templating software. Freehill and colleagues[1] evaluated the accuracy of implant size selection based on 3-D templating using both stemmed and stemless implants. The authors found that virtual pre-operative planning can reliably predict the component size utilized for anatomic total shoulder arthroplasty, though the stemless system had a greater predicted accuracy for the humeral component (98% matched the pre-operative template within one size with 79% exactly matched) when compared to the stemmed system (88% of cases were within one size of the preoperative plan and exactly matching in 83%). The accuracy for the humeral head sizes on the stemmed component were also overall lower with 79% matching within one size compared to 97% with a stemless humeral implant.  Statistical differences between stemmed and stemmed groups for pre-operative 3-D accuracy were not analysed.  Enhanced accuracy of 3-D pre-operative planning may reduce implant costs, operative times and inventory processes related to shoulder arthroplasty[1].  More research is required to determine the effect of accurate pre-operative 3-D technologies on clinical outcomes in shoulder arthroplasty, both for stemmed and stemless systems[2].

The conclusion supports the study.

The bibliography is appropriate but can be extended as suggested above.

The reference list has been extended with updated references.

Reviewer 4 Report

The authors present medium-term clinical outcomes of 30 60 total shoulder replacements implanted for glenohumeral osteoarthritis using the Global Icon stemless shoulder system, with a follow-up period of 24-months. There was a significant improvement at 24 months on the OSS and the WOOS Index. No component significantly loosened, migrated, or subsided. No implant related complications were recorded. The median operative time was only 75.5 minutes. The authors conclude that their study on the Global Icon system is at least comparable if not favourable when compared to other published studies on stemless implants.

Some comments:

2.2.

For those less familiar with the subject, it is difficult to get clarity about the surgical technique when talking about the glenoid peg in connection with the humeral component. I recommend adding a couple of sentences about the treatment of the glenoid side as well.

Line 119

The authors should also tell the minimum detectable difference (MDD) of those scores if one exists.

Line 332

Please add: 30% dropped out of the study, causing a significant possibility of error.

Fig. 1 is very dark. Is it possible to lighten it to improve clarity.

Author Response

Reviewer 4

The authors present medium-term clinical outcomes of 30 60 total shoulder replacements implanted for glenohumeral osteoarthritis using the Global Icon stemless shoulder system, with a follow-up period of 24-months. There was a significant improvement at 24 months on the OSS and the WOOS Index. No component significantly loosened, migrated, or subsided. No implant related complications were recorded. The median operative time was only 75.5 minutes. The authors conclude that their study on the Global Icon system is at least comparable if not favourable when compared to other published studies on stemless implants.

Some comments:

2.2.

For those less familiar with the subject, it is difficult to get clarity about the surgical technique when talking about the glenoid peg in connection with the humeral component. I recommend adding a couple of sentences about the treatment of the glenoid side as well.

We thank the reviewer for this comment.  We have updated the following paragraph to include more detail on the anchor-peg.

Line 85

An Anchor Peg Glenoid (DePuy Synthes, Warsaw, IN, USA) was implanted with cement in the peripheral holes only for immediate fixation (and none on the under surface of the component), and cancellous bone autograft reamings applied circumferentially around the central interference peg for bony ongrowth, as previously published in short and longer term outcome studies using this implant.[3, 4](Figure 1). The Global Icon humeral head was sized off the resected humeral head and implanted with the base plate gaining excellent metaphyseal bone contact and morse taper connection to the head.

The subscapularis tenotomy was subsequently repaired end-to-end with a contracting strong, synthetic suture in an interrupted fashion (Dynacord ®). A standardized rehabilitation protocol was utilized with shoulder pendulum exercises in the first 2 weeks after surgery

Line 119

The authors should also tell the minimum detectable difference (MDD) of those scores if one exists.

We have added the following to line 125

To date, no studies have reported a minimally clinically important difference (MCID) value for the WOOS Index, while the MCID for the OSS is 6.3 points[5].

Line 332

Please add: 30% dropped out of the study, causing a significant possibility of error.

The authors thank the reviewer for this comment.  We have now added the following to the manuscript:

Line 349

In addition, of the 43 patients retrieved in our data base, 30 had outcomes available for analysis resulting in possible measurement error

Fig. 1 is very dark. Is it possible to lighten it to improve clarity.

Figure 1 has been lightened.